# REDUCING SOURCE-PRIVATE BIAS IN EXTREME UNIVERSAL DOMAIN ADAPTATION

## ABSTRACT

Universal Domain Adaptation (UniDA) aims to transfer knowledge from a labeled source domain to an unlabeled target domain without prior knowledge of the label sets between the two domains. The goal of UniDA is to achieve robust performance under arbitrary label-set distributions. However, existing literature has not sufficiently explored performance across diverse distribution scenarios. Our experiments reveal that existing methods struggle when the source domain has significantly more non-overlapping classes than overlapping ones, a setting we refer to as *Extreme UniDA*. In this paper, we demonstrate that classical partial domain alignment, which focuses on aligning only overlapping-class data between domains, is limited in mitigating bias toward source-private classes in extreme UniDA scenarios. We argue that feature extractors trained with source supervised loss disrupt the intrinsic structure of target data due to the inherent differences between source-private-class data and target data. To mitigate this bias, we employ self-supervised learning to preserve the structure of target data. This method can be easily integrated into existing frameworks. We apply the proposed approach to two distinct training paradigms—adversarial-based and optimal-transport-based—and show consistent improvements across various class-set distributions, with significant gains in extreme UniDA settings.

## 1 INTRODUCTION

Advancements in deep learning and machine learning often rely on the assumption that abundant data follows the same distribution, making it difficult for these models to generalize well on unseen data sampled from different distributions. Unsupervised Domain Adaptation (UDA) (Pan & Yang, 2009) addresses this issue by transferring knowledge from a source domain with a known distribution to a target domain with a possibly different distribution. Nevertheless, most UDA methods (Ganin et al., 2016; Long et al., 2018; Jiang et al., 2020) for multi-class classification operate under the strong assumption that the label sets of the source domain ($\mathcal{C}_s$) and the target domain ($\mathcal{C}_t$) are the same ($\mathcal{C}_s = \mathcal{C}_t$), limiting the applicability in real-world scenarios.

To overcome this limit, more flexible setups such as Open-set Domain Adaptation ($\mathcal{C}_s \subset \mathcal{C}_t$) and Partial Domain Adaptation ($\mathcal{C}_s \supset \mathcal{C}_t$) have been studied (Panareda Busto & Gall, 2017; Saito et al., 2018; Cao et al., 2018b), addressing scenarios where the target label set has more or fewer classes than the source label set. Universal Domain Adaptation (UniDA) further loosens the setup by not assuming $\mathcal{C}_s$ and $\mathcal{C}_t$ to have any containment relation. Instead, UniDA allows for overlap between $\mathcal{C}_s$ and $\mathcal{C}_t$ on some unknown shared classes, while each set may also contain private, non-overlapping classes. The objective of UniDA is to classify target examples either as belonging to one of the shared classes or as an out-of-source class.

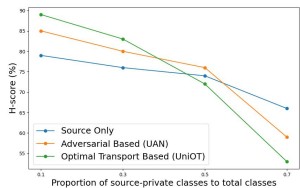

Figure 1: Comparison of prior works with training with *source data only* on Office31.

The ultimate goal of UniDA is to achieve robust performance regardless of label-set distributions. Nevertheless, prior works have mainly adhered to the experimental protocols established by You et al. (2019) and Fu et al. (2020), which have not comprehensively explored various distribution scenarios. In our thorough analysis in Figure 1, we identify a critical challenge: prior works fail

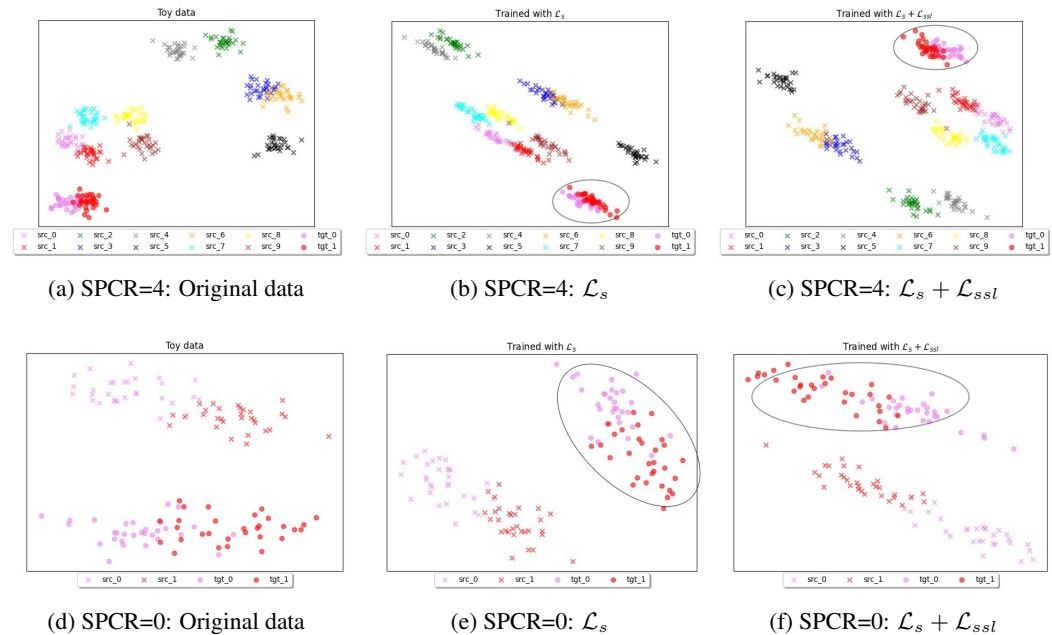

Figure 2: **Toy Example Visualizations under Different SPCR** ($|\overline{\mathcal{C}}_s|/|\mathcal{C}|$). Rows correspond to SPCR of 4 or 0, simulating scenarios with significant or minimal source-private classes, respectively. Columns show (a)(d) original data, (b)(e) features trained with supervised loss on source data ($\mathcal{L}_s$), and (c)(f) features trained with $\mathcal{L}_s$ and self-supervised loss on target data ($\mathcal{L}_{ssl}$). For SPCR = 4, training solely with $\mathcal{L}_s$ (b) causes notable distortion in target representations, while SPCR = 0 (e) shows minimal distortion. Adding $\mathcal{L}_{ssl}$ (c) reduces distortion and better preserves the data structure. Colors denote classes, with circles for source and crosses for target data. Further details are provided in Section 3.3

to address cases where the number of source-private classes significantly exceeds the number of common classes—a challenging sub-task we define as *Extreme UniDA*. This observation motivates us to investigate why prior works fall short in this sub-task.

We start by examining the typical process of solving UniDA. The process begins with training the feature extractor on the source data only. Then, a domain alignment loss, such as adversarial loss (Ganin et al., 2016; Long et al., 2018; Jiang et al., 2020) or self-training (Mei et al., 2020; Liu et al., 2021), is applied to align the source and target feature distributions. However, due to the presence of private data, directly aligning these distributions can lead to significant bias (e.g., target-common data aligning with source-private data). To mitigate this, PADA (Cao et al., 2018b) and UAN (You et al., 2019) initiated the concept of *partial domain alignment*, which designs weighting functions to downweight private-class data and focus alignment only on common-class data. Building on this idea, subsequent works (Liu et al., 2019; Lifshitz & Wolf, 2021; Saito et al., 2020; Chen et al., 2022) have explored more effective designs for weighting functions.

In Section 2, we demonstrate that partial domain alignment faces significant limitations in extreme UniDA scenarios. As illustrated in Figure 3, partial domain alignment must be highly precise to mitigate the bias towards source-private classes. However, achieving such accuracy is unrealistic without access to target labels. In this work, we explore the question:

*What is the gap that leads to source-private bias under Extreme UniDA?*

In Section 3.3, we present a toy analysis demonstrating that the feature extractor, when trained with source supervised loss, focus on learning the direction for classifying source data, thereby neglecting the direction relevant to target data. The effect of this phenomenon is amplified in Extreme UniDA scenarios (Figure 2b, 2e), where the intrinsic spatial differences between source-private data and target data become increasingly pronounced. To address this gap, we propose leveraging the capabilities of self-supervised learning (SSL) to preserve the structure of target data. We systematically

investigate how applying SSL to target-private classes influences results and its compatibility with partial domain alignment. In addition, our toy experiment shows that SSL can maintain the direction for target data (Figure 2c), effectively mitigating source-private bias and allow subsequent domain alignment loss to function more effectively. This finding is further validated in Section 4.3 through singular value spectrum analysis on real datasets. It further elucidates its application in the domain adaptation literature and explains why SSL performs particularly well in Extreme UniDA scenarios, a context that has not been thoroughly explored in prior works (Xu et al., 2019; Bucci et al., 2019; 2021; Achituve et al., 2021).

Our proposed self-supervised loss specifically aims to reduce source-private bias, providing an approach that is orthogonal to previous UniDA methods. This loss is lightweight and can be easily integrated as an add-on module to existing frameworks. We apply the proposed loss in two distinct training paradigms: adversarial-based methods and optimal transport-based methods (Section 3.2). Our results across multiple universal domain adaptation datasets demonstrate improved model robustness across various class set distributions, particularly in extreme UniDA scenarios (Section 4.2).

Our contributions can be summarized as follows:

1. We are the first to investigate the unsolved *Extreme UniDA* problem, highlighting the limitations of current UniDA methods.

2. We provide a deeper understanding that the widely used partial domain alignment paradigm fails when when the amount of source-private data is large.

3. We propose incorporating target label information by SSL as a lightweight module for partial domain alignment, which can reduce source-private bias and significantly enhance robustness across varying class-set distributions.

4. We are the first to systematically explore various aspects of applying SSL to UniDA, including the impact of target-private classes, the severity of source-private bias, and benefits of combining SSL with partial domain alignment.

## 2 UNDERSTANDING THE LIMITATION OF PARTIAL DOMAIN ALIGNMENT

Universal Domain Adaptation (UniDA) (You et al., 2019) has a labeled source domain $\mathcal{D}_s = \{(\mathbf{x}_i^s, y_i^s)\}_{i=1}^{n_s}$ and an unlabeled target domain $\mathcal{D}_t = \{\mathbf{x}_i^t\}_{i=1}^{n_t}$ accessible at training time. The datasets $\mathcal{D}_s$ and $\mathcal{D}_t$ are sampled from the source and target distributions $p_s$ and $p_t$, respectively. The label sets are denoted as $\mathcal{C}_s$ for the source domain and $\mathcal{C}_t$ for the target domain, respectively. $\mathcal{C} = \mathcal{C}_s \cap \mathcal{C}_t$ represents the common label set shared by both domains. Let $\overline{\mathcal{C}}_s = \mathcal{C}_s \setminus \mathcal{C}$ and $\overline{\mathcal{C}}_t = \mathcal{C}_t \setminus \mathcal{C}$ represent the source-private label set and the target-private label set, that a label only occurs at one of two domains. The goal of UniDA is to classify target examples into $|\mathcal{C}| + 1$ classes, where target-private examples are regarded as one unknown class.

In this work, we propose Source-Private to Source-Common Ratio (SPCR) to systemically study the UniDA under various label-set distributions. SPCR is defined as the ratio of the number of source-private classes to the number of source-common classes $\frac{|\overline{\mathcal{C}}_s|}{|\mathcal{C}|}$. Intuitively, *a high SPCR may cause the model prediction to become biased towards source-private classes*. Throughout the paper, we utilize this ratio to analyze how partial domain alignment performs across different UniDA scenarios.

In Figure 1, we observe that the widely-used partial domain alignment framework performs well in settings with low SPCR but fails to outperform the `source-only` baseline in settings with high SPCR. To address this, we provide a detailed analysis explaining why this occurs. In Section 2.1, we begin by introducing the commonly adopted partial domain alignment approach in the UniDA literature, which aims to reduce source-private bias by aligning only the common-class data across domains. This method has shown promising results in general UniDA settings with low SPCR. In Section 2.2, we examine its limitations, exploring both *why it performs well in low SPCR scenarios* and *why it struggles in high SPCR cases*.

## 2.1 BACKGROUND: PARTIAL DOMAIN ALIGNMENT

The adversarial-based partial domain alignment is commonly used in universal and partial domain adaptation (Cao et al., 2018b; Liu et al., 2019; You et al., 2019; Fu et al., 2020; Lifshitz & Wolf, 2021), which consists of a feature extractor $\theta_f$, a label classifier $\theta_c$, and a domain discriminator $\theta_d$. The adversarial-based partial domain alignment includes cross-entropy loss and adversarial alignment loss. The cross-entropy trains a label classifier $\theta_c$ using labeled source data:

$$\mathcal{L}_s(\theta_f, \theta_c) = \mathbb{E}_{(\mathbf{x},y)\sim p_s} \mathbb{CE}(y, \theta_c(\theta_f(\mathbf{x}))). \tag{1}$$

Given the label-set shift between the source data and target data, predictions generated by models trained with $\mathcal{L}_s$ may become biased towards source-private classes. This bias is further exacerbated by the distribution shift, which increases the statistical likelihood of misclassification in the target data.

To reduce this bias, previous works have introduced partial domain alignment, which focuses on aligning only the common-class samples between domains. This can be incorporated into the adversarial alignment loss as follows:

$$\mathcal{L}_{adv}(\theta_f, \theta_d) = -\mathbb{E}_{\mathbf{x}\sim p_s} w_s(\mathbf{x}) \log \theta_d(\theta_f(\mathbf{x})) - \mathbb{E}_{\mathbf{x}\sim p_t} w_t(\mathbf{x}) \log(1 - \theta_d(\theta_f(\mathbf{x}))) \tag{2}$$

Here, $w_s(\mathbf{x})$ and $w_t(\mathbf{x})$ are used to downweight private-class samples from both domains to ensure that only common-class samples are aligned during training. The overall objective can be formulated as:

$$\min_{\theta_f, \theta_c} \max_{\theta_d} \big[ \mathcal{L}_s(\theta_f, \theta_c) - \lambda \mathcal{L}_{adv}(\theta_f, \theta_d) \big], \tag{3}$$

where $\lambda$ is the weighted hyperparameter.

**The role of $w_s(\mathbf{x})$ and $w_t(\mathbf{x})$.** As shown in Equation 2, effectively reducing bias depends on perfect partial alignment, i.e., accurately approximating $w_s(\mathbf{x})$ and $w_t(\mathbf{x})$. Prior research has explored various uncertainty metrics to calculate these weights. For instance, Cao et al. (2018a) employs class probability, while Saito et al. (2018); Cao et al. (2018b) utilize confidence scores, and You et al. (2019) leverage entropy. Additionally, Zhang et al. (2018) use domain similarity calculated with an auxiliary domain discriminator. There are also approaches that combine multiple uncertainty metrics into an ensemble (Lifshitz & Wolf, 2021; Fu et al., 2020). These methods have shown promising results in general UniDA settings with low SPCR. We provide a detailed explanation of the uncertainty measurements in Appendix B.5.

## 2.2 LIMITATION OF PARTIAL DOMAIN ALIGNMENT

Section 2.1 presents the goal of partial domain alignment, which is to reduce source-private bias. Although partial domain alignment achieves promising results across various label set distributions, it struggles under high SPCR, as demonstrated in Figure 1. In such cases, partial domain alignment ($\mathcal{L}_s + \mathcal{L}_{adv}$) performs worse than using the source loss only ($\mathcal{L}_s$). To investigate the underlying cause of this failure, we aim to *evaluate the effectiveness of partial domain alignment* and assess *whether it can mitigate bias across all conditions*.

**Evaluation of $w_s(\mathbf{x})$ and $w_t(\mathbf{x})$.** We first need an evaluation method to evaluate the effectiveness of partial domain alignment. In an ideal case, the alignment weight $w_s(\mathbf{x})$ and $w_t(\mathbf{x})$ should assign 0 to private-class samples and 1 to common-class samples. In other words, the data used for partial domain alignment should ideally contain only common-class samples. When private-class samples are included, they are treated as "noise" in the alignment process. We define the noise rate of a batch $B$ for partial domain alignment as:

$$\frac{1}{|B|} \sum_{(\mathbf{x},y)\in B} \mathbb{I}\{\hat{y}(\mathbf{x}) \neq \mathbb{I}\{y \in \mathcal{C}\}\}, \tag{4}$$

where $\hat{y}(\mathbf{x}) = \mathbb{I}\{w_\bullet(\mathbf{x}) \geq 0.5\}$. With this metric, we can explore how much noise in partial domain alignment can be tolerated while still achieving better performance than training with source data only.

We begin by evaluating the performance under SPCR set to 2 to investigate the reason behind the success of partial domain alignment in previous works. To do this, we manually increase the noise rate in partial domain alignment (details in Appendix B.6) and observe at what point it begins to underperform compared to the `source-only` baseline. As shown in Figure 3a, the performance of partial domain alignment only starts to decline when the noise rate exceeds 0.35. Figure 3b demonstrates that different partial domain alignment methods have noise rates of around 0.25-0.3, less than the tolerance noise rate of 0.35. These results explain the effectiveness of partial domain alignment in reducing bias in UniDA.

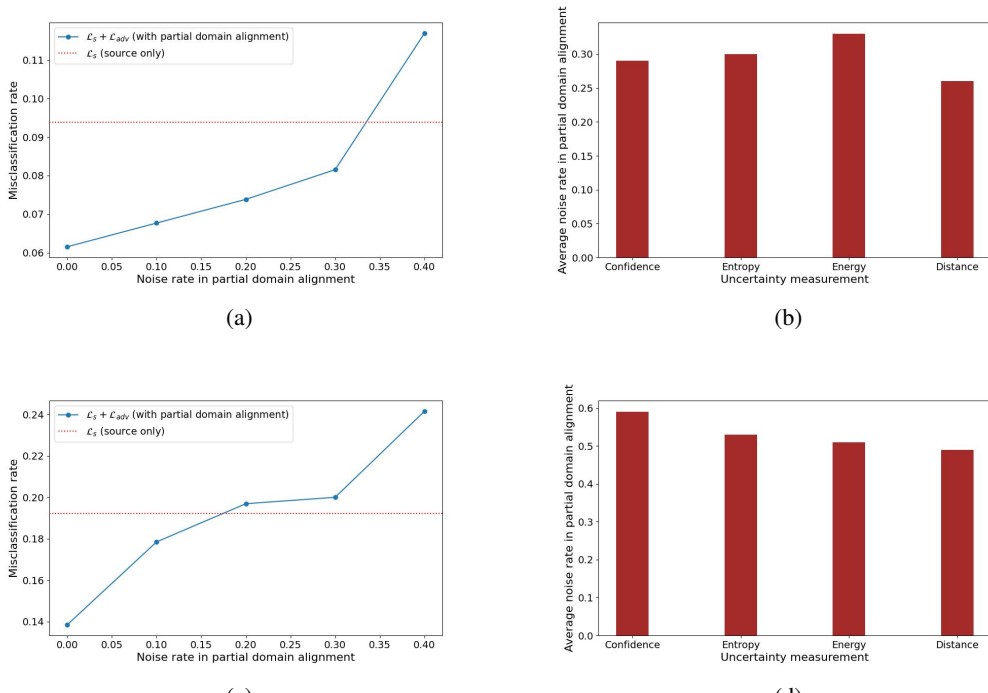

Figure 3: **Tolerance of Noise Rate in Partial Domain Alignment**: (a) (c) The misclassification rate under different noise levels for SPCR=2 and 5, respectively. (b) (d) The observed noise rate using common uncertainty measurements to down-weight private samples, averaged across all batches for SPCR=2 and 5, respectively. The misclassification rate specifically refers to the misclassification of target common-class data into source-private classes, evaluating *the prediction bias toward source-private classes*. Results are reported on OfficeHome.

Next, we increase SPCR to 5 and observe a different trend. In contrast to previous results, the tolerance noise rate decreases to 0.2 in Figure 3c and the average noise rate in existing partial domain alignment methods are way much higher than the tolerance noise rate in Figure 3d. These findings suggest that partial domain alignment exacerbates performance by introducing excessive noise, amplifying the bias, which in turn increases the noise further, creating a vicious cycle.

## 3 METHODOLOGY: REDUCING SOURCE-PRIVATE BIAS WITH SELF-SUPERVISED LEARNING

### 3.1 AN ALTERNATIVE WAY TO REDUCE SOURCE-PRIVATE BIAS

As discussed in Section 2.2, partial domain alignment fails to effectively reduce bias under Extreme UniDA. The high bias leads to low tolerance for noise, while the imprecise weights tend to exhibit high levels of noise in partial domain alignment. To break the vicious cycle, we pose the question: *Are there alternative methods to reduce source-private bias?* We first delve into the property of Extreme UniDA. In this setting, the presence of many source-private classes biases the source-

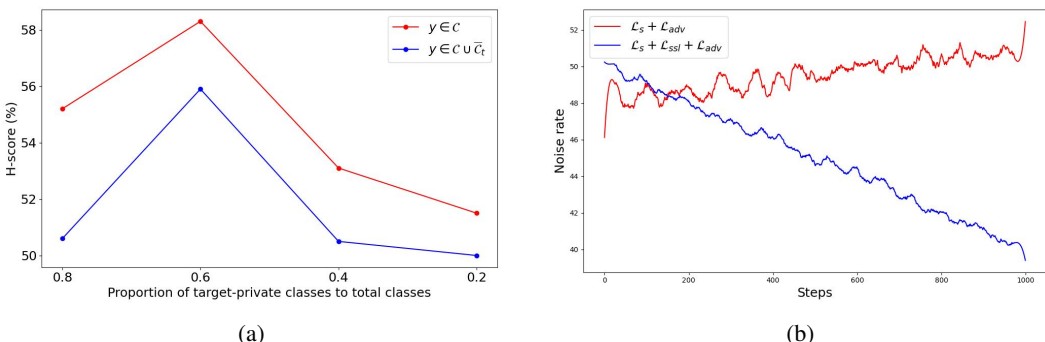

(a)                                                            (b)

Figure 4: (a) Performance comparison of applying SSL on target data: common classes vs. all target classes under varying proportions of target-private classes. The experiments are conducted on three settings of Office-Home. (b) The noise rate of partial domain alignment with and without SSL.

supervised loss heavily towards these classes, often leading to poor performance on target classes. Therefore, we argue that a method to reduce source-private bias must be applied concurrently with the source-supervised loss. In contrast to partial domain alignment, which seeks to mitigate bias by removing source-private information, we turn to the novel approach incorporating target data to address bias. This requires careful incorporation of target data with minimal reliance on source data, given the lack of target domain labels and the significant presence of source private-class data. Self-training (Mei et al., 2020; Liu et al., 2021) is a popular method for incorporating target data by selecting high-confidence samples as pseudo-labels for further training. However, it remains heavily reliant on source data, which makes it vulnerable to source-private bias. In this work, we leverage self-supervised learning to learn from unlabeled data, as it eliminates the dependence on source data and effectively captures the underlying structure of the target data.

The self-supervised loss is formulated as:

$$\mathcal{L}_{ssl}(\theta_f) = \mathbb{E}_{\mathbf{x} \sim p_t} ||\theta_f(\mathcal{T}(\mathbf{x})) - \theta_f(\mathcal{T}'(\mathbf{x}))||^2, \tag{5}$$

where $\mathcal{T}$ and $\mathcal{T}'$ are independent random augmentation functions.

While SSL has been applied to DA-related tasks (Xu et al., 2019; Bucci et al., 2019; 2021; Achituve et al., 2021; Saito et al., 2020), several questions remain to be explored: (1) Does applying SSL to target private-class data harm performance? (2) Why can SSL be effectively incorporated into partial domain alignment? We address (1) in this section and (2) in Section 3.2. To explore (1), we conducted an ablation study by applying SSL exclusively on the target common-class data and compared it to applying SSL on the entire target dataset. The results in Figure 4a indicate that including target-private-class data does indeed hurt performance, particularly when there is a high proportion of target-private classes. However, the performance decline is relatively minor compared to the benefits it brings (as shown in Section 4). We argue that SSL focuses on learning data structure rather than classifying target data, as supervised loss does, which makes the performance drop less severe compared to that introduced by supervised learning.

## 3.2 A UNIFIED FRAMEWORK WITH SELF-SUPERVISED LEARNING

In this section, we demonstrate that SSL can be effectively integrated into the partial domain alignment framework, yielding positive results. Although previous works (Bucci et al., 2019; 2021) have combined SSL with adversarial domain alignment in other domain adaptation tasks, there has been no investigation into their complementary effects. We aim to discuss *how SSL enhance partial domain alignment*.

We conduct an experiment comparing the average noise rate of partial domain alignment with and without SSL. As shown in Figure 4b, training with SSL significantly reduces the noise rate in partial domain alignment. In other words, SSL helps partial domain alignment break the vicious cycle of error accumulation, further improving performance. These experimental results support our decision to unify the two approaches.

Finally, we design the unified framework to combine partial domain alignment with self-supervised learning:

$$\min_{\theta_f, \theta_c} \max_{\theta_d} \left[ \mathcal{L}_s(\theta_f, \theta_c) - \lambda \mathcal{L}_{adv}(\theta_f, \theta_d) + \alpha \mathcal{L}_{ssl}(\theta_f) \right], \tag{6}$$

where $\lambda$ and $\alpha$ are the weighted hyperparameters. While we use adversarial-based methods as an example, the SSL loss term $\mathcal{L}_{ssl}$ can also be applied to other domain alignment methods, such as optimal transport (Chang et al., 2022), by replacing $\mathcal{L}_{adv}$ with the corresponding alignment loss and removing the max term.

### 3.3 A STUDY OF SSL WITH TOY EXPERIMENTS

In this section, we present an intuitive experiment to illustrate how source-private bias forms in UniDA, why it intensifies as SPCR increases and how SSL can effectively mitigate the bias. We argue that this bias emerges during the training of the feature extractor using $\mathcal{L}_s$. To support this claim, we present a toy experiment that illustrates how target features shift when the model is trained exclusively on source data. In Section 4.3, we further validate these observations using the singular value spectrum on a real dataset.

**Toy dataset construction**  Motivated by Liu et al. (2022), we generate a 2D toy dataset under the framework of universal domain adaptation as illustrated in Figure 2a, 2d. Let $e_1$ and $e_2$ be two orthogonal unit vectors in $\mathbb{R}^2$. The source data is generated as $\mathbf{x}_s = \tau e_1 + \gamma e_2 + \epsilon$, where $\tau, \gamma > 0$ are hyperparameters controlling the positions of the class centroids, and $\epsilon \sim \mathcal{N}(0, I)$. To simulate the distribution shift in $p(\mathbf{x})$ of target-common classes for domain adaptation problem, we apply a rotation matrix $R(\theta)$ to the source-common-class data. Specifically, the target data is given by $\mathbf{x}_t = R(\theta)\mathbf{x}_s$, where $R(\theta)$ is a 2D rotation matrix parameterized by angel $\theta$. The rotation is chosen such that the target data $\mathbf{x}_t$ aligns predominantly along the direction $e_1$, i.e., $\mathbf{x}_t = \rho e_1$, where $\rho$ is a scalar. For clarity in visualization, we do not generate target-private-class data, as its absence does not affect our demonstration.

**Method formulation**  We consider a two-layer linear network with ReLU as the activation function (Agarap, 2018). For supervised learning, we minimize the objective: $\mathcal{L}_s(W_1, W_2) = \mathbb{E}_{(\mathbf{x},y) \sim \mathcal{D}_s} ||W_2(W_1(\mathbf{x})) - y||^2$. For self-supervised learning, similar to SimSiam (Chen & He, 2021), we minimize the objective $\mathcal{L}_{ssl}(W_1) = \mathbb{E}_{\mathbf{x} \sim \mathcal{D}_t} ||W_1(\mathbf{x} + \epsilon) - W_1(\mathbf{x} + \epsilon')||^2$, where $\epsilon$ and $\epsilon'$ are random perturbations. We compare models trained using $\mathcal{L}_s$ alone with those trained with $\mathcal{L}_s + \mathcal{L}_{ssl}$.

**Results**  As shown in Figure 2b, training with only the source loss $\mathcal{L}_s$ leads to learning a direction that is a linear combination of $e_1$ and $e_2$. This cause the target features to deform or misalign, as the model fails to capture the specific direction of $e_1$ in the target data. This observation highlights the limitations of partial domain alignment in correcting the biases in the feature extractor. When a significant number of source-private classes exist, the source loss focuses on learning directions that discriminate these source-private classes. This focus can lead to poor generalization to the target domain, requiring near-perfect partial domain alignment to correct such bias.

In contrast, as shown in Figure 2c, when training with $\mathcal{L}_s + \mathcal{L}_{ssl}$, we can observe the target features maintain its direction along $e_1$. The observations motivates us to apply SSL on extreme UniDA, where the feature extractor may exhibit significant bias.

## 4 EXPERIMENTS

### 4.1 EXPERIMENTAL SETTINGS

**Dataset.**  We present results on four widely used benchmarks: Office31, OfficeHome, VisDA, and DomainNet. Detailed descriptions of these datasets can be found in Appendix B.3.

**Extreme UniDA setting.**  Previous studies have explored various class-set distributions, but they are constrained to settings with SPCR $< 1$, limiting the evaluation of models' robustness under diverse class-set distributions. To address this, we introduce settings with SPCR $> 1$ to evaluate

Table 1: H-score (%, ↑) on **Office-Home** and **VisDA**. For each column, the best values are high-lighted in **bold**, while the top value in each category is highlighted with underline.

| | Office-Home | | | | | | | | | | | | VisDA |
| | Ar2Cl | Ar2Pr | Ar2Rw | Cl2Ar | Cl2Pr | Cl2Rw | Pr2Ar | Pr2Cl | Pr2Rw | Rw2Ar | Rw2Cl | Rw2Pr | Avg | S2R |
|---|---|---|---|---|---|---|---|---|---|---|---|---|---|---|
| **Adversarial-based** | | | | | | | | | | | | | | |
| UAN (You et al., 2019) | 29.9 | 36.4 | 14.1 | 22.2 | 20.6 | 16.4 | 26.4 | 25.1 | 27.3 | 31.3 | 24.4 | 35.4 | 25.8 | 41.5 |
| CMU (Fu et al., 2020) | 38.5 | 43.5 | 45.7 | 41.4 | 41.2 | 47.5 | 46.0 | 46.6 | 40.3 | 41.5 | 38.5 | 27.2 | 41.5 | 34.1 |
| DANCE (Saito et al., 2020) | 34.0 | 55.5 | **82.6** | 43.4 | 44.2 | 60.1 | 34.4 | 20.8 | 61.2 | 65.7 | 33.6 | 61.7 | 44.6 | 69.1 |
| **UAN+SSL** | 47.1 | **74.4** | 76.8 | 46.4 | 54.3 | 63.5 | 55.8 | 48.5 | **72.3** | 57.7 | 46.3 | **68.5** | 59.3 | 89.5 |
| **OT-based** | | | | | | | | | | | | | | |
| UniOT (Chang et al., 2022) | 27.2 | 32.3 | 26.6 | 28.4 | 29.9 | 23.2 | 31.4 | 29.3 | 23.0 | 35.9 | 34.3 | 35.3 | 29.7 | 49.9 |
| **UniOT+ SSL** | 32.1 | 30.3 | 31.0 | 29.7 | 28.9 | 25.6 | 32.1 | 33.9 | 29.1 | 36.7 | 35.3 | 45.6 | 32.5 | 61.1 |
| **Others** | | | | | | | | | | | | | | |
| MLNet (Lu et al., 2024) | 58.2 | 66.5 | 63.3 | 69.4 | 71.2 | 64.1 | 51.3 | 59.6 | 67.7 | 49.9 | **65.3** | 56.3 | 61.9 | 75.1 |
| **MLNet+ SSL** | **59.4** | 71.5 | 72.9 | **71.0** | **71.5** | **66.7** | **57.5** | **60.4** | 68.8 | 59.9 | 64.4 | 53.3 | **64.8** | 80.2 |

Table 2: H-score (%, ↑) on **Office** and **DomainNet**. For each column, the best values are highlighted in **bold**, while the top value in each category is highlighted with underline.

| | Office | | | | | | | DomainNet | | | | | | |
| | A2D | A2W | D2A | D2W | W2A | W2D | Avg | P2R | R2P | P2S | S2P | R2S | S2R | Avg |
|---|---|---|---|---|---|---|---|---|---|---|---|---|---|---|
| **Adversarial-based** | | | | | | | | | | | | | | |
| UAN (You et al., 2019) | 24.5 | 61.8 | 48.9 | 64.2 | 27.9 | 61.3 | 48.1 | 11.9 | 15.1 | 14.4 | 17.2 | 18.1 | 11.3 | 14.6 |
| CMU (Fu et al., 2020) | 76.8 | 63.8 | 56.1 | 77.2 | 66.3 | 78.2 | 69.7 | 30.1 | 42.4 | 34.1 | 24.3 | 32.2 | 34.1 | 32.8 |
| DANCE (Saito et al., 2020) | 49.7 | 47.9 | 48.4 | 54.9 | 48.9 | 55.6 | 50.9 | 39.4 | 3.30 | 11.8 | 0.90 | 7.60 | 35.3 | 16.4 |
| **UAN+ SSL** | **87.4** | 74.9 | 72.4 | **81.3** | 74.9 | **87.7** | 79.8 | **50.1** | 39.2 | 35.9 | 32.7 | 34.0 | 49.8 | 40.3 |
| **OT-based** | | | | | | | | | | | | | | |
| UniOT (Chang et al., 2022) | 78.8 | 67.7 | **86.1** | 66.9 | 83.8 | 81.0 | 77.4 | 38.1 | 29.8 | 30.8 | 29.3 | 29.1 | 38.3 | 32.6 |
| **UniOT+ SSL** | 79.8 | 75.9 | 86.0 | 77.3 | **84.1** | 82.4 | **80.9** | 39.6 | 29.9 | 33.6 | 31.4 | 31.1 | 40.2 | 34.3 |
| **Others** | | | | | | | | | | | | | | |
| MLNet (Lu et al., 2024) | 51.2 | 61.9 | 58.1 | 79.2 | 59.5 | 75.5 | 64.2 | 48.0 | 45.9 | 47.8 | 48.5 | 43.7 | **53.0** | 47.8 |
| **MLNet+ SSL** | 48.6 | 60.5 | 60.2 | 80.6 | 60.7 | 80.3 | **65.2** | 48.8 | **46.5** | **50.2** | **50.6** | **44.7** | 52.5 | **48.9** |

models in scenarios where source-private bias is more severed, as shown in Table 6. Due the space limitation, we put the results of general UniDA setting in Appendix A.1.

**Evaluation metric.** We adopt the widely used H-score (Fu et al., 2020) as the metric, which calculate the harmonic mean of accuracy on common classes $a_{\mathcal{C}}$ and accuracy on target-private (unknown) classes $a_{\overline{\mathcal{C}}_t}$. See Appendix B.2 for details.

**Baselines.** For the baselines, we considered methods that incorporate domain alignment loss and have open-source codebases. We categorized them into two groups: adversarial-based and optimal-transport-based methods. In the adversarial-based category, we included UAN, CMU, and DANCE. For the optimal-transport-based category, we used the only existing work, UniOT.

**Implementation details.** See Appendix B.4 for details.

## 4.2 MAIN RESULTS ON EXTREME AND GENERAL UNIDA SETTING

Table 1 and 2 summarize the results in the extreme UniDA setting across four different DA benchmarks. We demonstrate that combining SSL with adversarial-based methods yields significant improvements. Specifically, it outperforms the best adversarial-based method, CMU, by 10.1% on Office, 7.5% on DomainNet, and 17.8% on Office-Home, while surpassing DANCE by 20.4% on VisDA. When applied to OT-based methods, although the improvements are less pronounced, the results consistently show gains over UniOT, with increases of 2.8% on Office-Home, 11.2% on VisDA, 3.5% on Office, and 1.7% on DomainNet.

Results of the general UniDA setting with low SPCR are provided in Appendix A.1. These results indicate that the improvements are relatively modest compared to the extreme setting, with gains of 8.6% on Office-Home and 15.5% on Office for the adversarial-based method, and 0.5% on Office-Home and 0.4% on Office for the OT-based method. This observation aligns with our hypothesis that

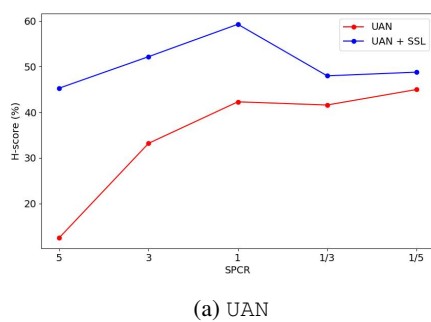 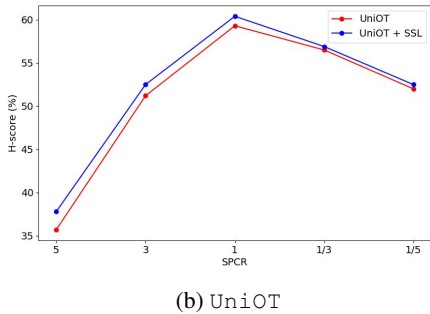

(a) UAN                                   (b) UniOT

Figure 5: **Comparison of baseline models with and without SSL across various label-set distributions.** The results are evaluated on DomainNet.

SSL is more effective in the extreme setting with high SPCR. In summary, SSL brings significant improvements in the extreme UniDA setting and has marginal gains in the general UniDA setting, demonstrating enhanced overall robustness across diverse SPCR levels.

### 4.3 DISCUSSION

We aim to answer the following questions: (1) Does adding the additional loss term ensure robust performance across varying label-set distributions? (2) What factors contribute to the performance gap in extreme UniDA scenarios? (3) What specific issues does SSL effectively resolve?

**Robustness to Varying Label-set Distributions.** While we introduce extreme settings to evaluate the model's performance with high SPCR, many intermediate ratios remain unexplored. To address this gap, we select the challenging DomainNet dataset to test the model across a range of SPCR values: $\{\frac{1}{5}, \frac{1}{3}, 1, 3, 5\}$. From Figure 5, it is clear that both UAN and UniOT benefit from SSL, highlighting its robustness across varying label-set distributions. Furthermore, the improvement becomes more pronounced as SPCR increases (though less significant for UniOT), suggesting that SSL is particularly effective in Extreme UniDA, where bias is more pronounced.

**Dimensional collapse under Extreme UniDA.** To validate our hypothesis that training with $\mathcal{L}_s$ in high SPCR settings leads to distortion of target representations, as shown in Figure 2, we plot the singular value spectrum of target representations in Figure 6a to assess representation quality. The results show that as SPCR increases, the number of singular values nearing zero also rises, indicating that target features are restricted to a low-dimensional subspace—a phenomenon known as dimensional collapse (Gao et al., 2019; Jing et al., 2022).

**SSL Prevents Dimensional Collapse** To test our hypothesis that incorporating $\mathcal{L}_{ssl}$ better preserves the structure of the target data, we analyzed the singular value spectrum before and after applying $\mathcal{L}_{ssl}$ in Figure 6b. The results demonstrate that SSL effectively mitigates dimensional collapse, highlighting its particular effectiveness in Extreme UniDA.

## 5 RELATED WORKS

**Self-supervised Learning for Domain Adaptation** Self-supervised learning (SSL) has been applied to various domain adaptation tasks, including unsupervised domain adaptation (Xu et al., 2019) using simple pretext tasks, partial domain adaptation (Bucci et al., 2019; 2021) with jigsaw puzzles, point cloud tasks (Achituve et al., 2021) involving deformation reconstruction, and universal domain adaptation through clustering based on source data (Saito et al., 2020). Our work extends this line of research by employing SSL for extreme UniDA. Unlike prior studies, our approach provides a deeper understanding of SSL's role in addressing source-private bias and its effectiveness in extreme UniDA scenarios—an aspect that has not been previously explored. Moreover, our method differs from DANCE (Saito et al., 2020), which relies on clustering using source data. In contrast, we propose leveraging target data independently of the source data. The advantage of our approach is

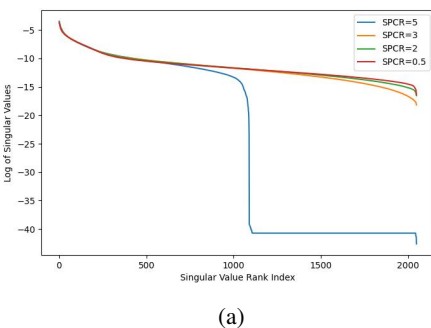 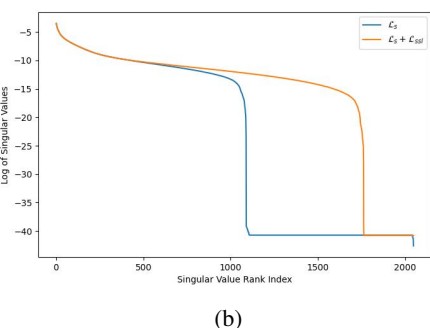

(a)                                                          (b)

Figure 6: **Singular Value Spectrum Analysis**: (a) The number of singular values approaching zero rises as SPCR increases. (b) Applying SSL can mitigate dimensional collapse. The results are conducted on OfficeHome (Pr2Cl), where the output dimension is 2048.

reflected in Table 1, where DANCE exhibits poor performance in extreme UniDA settings. Additionally, there are complementary studies highlighting the effectiveness of *pretraining* SSL in contexts of distribution shift or imbalanced learning. Garg et al. (2024) showed that combining self-training with contrastive learning pretraining outperforms either approach alone. Similarly, Liu et al. (2022) found that SSL can effectively learn the representations of minority classes, resulting in robust performance in imbalanced learning scenarios. Their toy experiments inspire the design of our own experiments, as discussed in Section 3.3.

**Universal Domain Adaptation** Universal domain adaptation is a more generalized form of unsupervised domain adaptation that makes no assumptions about the label sets relationship between the source and target domains. Numerous prior works (You et al., 2019; Fu et al., 2020; Lifshitz & Wolf, 2021; Saito et al., 2020; Chen et al., 2022) have focused on designing effective weighting functions to downweight the contribution of private samples in domain alignment. The design of these weighting functions is discussed in detail in Section 2.1. We cover these methods extensively in our paper as we found their limitations in addressing source-private bias. Another line of research (Saito & Saenko, 2021; Hur et al., 2023; Lu et al., 2024) focuses on designing robust open-set classifiers to distinguish between common classes and private classes in target data. Since these methods do not emphasize domain alignment, they are not covered in our paper. With the emergence of more advanced models, Zhu et al. (2023b); Deng & Jia (2023) explore the application of models such as vision transformers and pretrained vision models like DINO (Caron et al., 2021) and CLIP (Radford et al., 2021) to UniDA. There are also works that align with our goal of exploring more realistic or under-explored scenarios in UniDA. Qu et al. (2024) investigates source-free UniDA, where source data is unavailable during adaptation. Zhu et al. (2023a) addresses generalized UniDA, which aims to identify novel categories and label distributions in the target domain, utilizing active learning to achieve this objective.

## 6 CONCLUSION

We underline Extreme UniDA, a challenging sub-task of UniDA that remains unsolved and underexplored by existing UniDA methods. We argue that the difficulty of the task roots in the sourceprivate bias, and demonstrate that state-of-the-art UniDA methods, mostly designed by partial domain alignment that *removes* irrelevant data by reweighting, cannot completely mitigate the bias on their own for Extreme UniDA. The findings motivate us to devise a new methodology that works by *adding* relevant information to reduce the bias. The proposed methodology applies self-supervised learning to enrich the representation with the structural information of the source and target data. Our extensive experiments verify that the proposed methodology, albeit lightweight, effectively improves existing partial domain alignment methods across different ratios of source-private labels. In particular, the methodology achieves a significant gain when facing extreme UniDA scenarios. The promising results of our proposed methodology open a novel future research direction on how to *add* information systematically and strategically to improve UniDA methods.

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

## A   ADDITIONAL EXPERIMENTS

### A.1   RESULTS ON GENERAL UNIDA

Table 3 and 4 summarize the results for the general UniDA setting used in previous works. The results demonstrate that integrating SSL yields improvements in general UniDA as well, with gains of 0.4% on Office31 and 0.5% on Office-Home. While these improvements are marginal compared to those observed in the extreme UniDA setting, they support our hypothesis that SSL provides greater benefits in high SPCR scenarios. Moreover, the fact that SSL significantly improves performance in extreme UniDA without negatively affecting general UniDA highlights its overall contribution to robustness.

Table 3: H-score(%) on **Office** (10/10/11)

| | A2D | A2W | D2A | D2W | W2A | W2D | Avg |
|---|---|---|---|---|---|---|---|
| **Office (10/10/10)** | | | | | | | |
| **Adversarial-based** | | | | | | | |
| UAN (You et al., 2019) | 59.7 | 58.6 | 60.1 | 70.6 | 60.3 | 71.4 | 63.5 |
| CMU (Fu et al., 2020) | 68.1 | 67.3 | 71.4 | 79.3 | 72.2 | 80.4 | 73.1 |
| DANCE (Saito et al., 2020) | 72.6 | 62.4 | 63.3 | 76.3 | 57.4 | 82.8 | 66.6 |
| **UAN+ SSL** | 85.8 | 83.5 | 84.7 | 96.4 | 84.2 | **97.2** | 88.6 |
| **OT-based** | | | | | | | |
| UniOT (Chang et al., 2022) | **87.0** | 88.5 | 88.4 | **98.8** | 87.6 | 96.6 | 91.2 |
| **UniOT+ SSL** | 86.6 | **88.8** | **90.7** | 98.2 | **88.6** | 96.7 | **91.6** |

Table 4: H-score(%) on **Office-Home** (5/10/50) .

| | Ar2Cl | Ar2Pr | Ar2Rw | Cl2Ar | Cl2Pr | Cl2Rw | Pr2Ar | Pr2Cl | Pr2Rw | Rw2Ar | Rw2Cl | Rw2Pr | Avg |
|---|---|---|---|---|---|---|---|---|---|---|---|---|---|
| **Office-Home (5/10/50)** | | | | | | | | | | | | | |
| **Adversarial-based** | | | | | | | | | | | | | |
| UAN (You et al., 2019) | 51.6 | 51.7 | 54.3 | 61.7 | 57.6 | 61.9 | 50.4 | 47.6 | 61.5 | 62.9 | 52.6 | 65.2 | 56.6 |
| CMU (Fu et al., 2020) | 56 | 56.9 | 59.2 | 67.0 | 64.3 | 67.8 | 54.7 | 51.1 | 66.4 | 68.2 | 57.9 | 69.7 | 61.6 |
| DANCE (Saito et al., 2020) | 26.7 | 11.3 | 18.0 | 33.2 | 12.5 | 14.3 | 41.6 | 39.9 | 33.3 | 16.3 | 27.1 | 25.9 | 25.0 |
| **UAN+ SSL** | 53.8 | 75.1 | 83.9 | 63.2 | 67 | 77.6 | 72.2 | 55.9 | 81.6 | 74.2 | 55.9 | 81.6 | 70.2 |
| **OT-based** | | | | | | | | | | | | | |
| UniOT (Chang et al., 2022) | 67.3 | 80.5 | 86.0 | 73.5 | **77.3** | **84.3** | 75.5 | 63.3 | 86.0 | **77.8** | 65.4 | 81.9 | 76.6 |
| **UniOT+ SSL** | **70.1** | **80.7** | **87.3** | **73.8** | 76.7 | 84.0 | **76.1** | **63.9** | **86.2** | 77.4 | **66.3** | **83.1** | **77.1** |

### A.2   RESULTS WITH ERROR BAR

We report the results on Office31 (Saenko et al., 2010) based on three runs in Table 5, each using a different random seed. The standard deviation values are relatively minor compared to the advantages we observe over prior works. The results indicate that our method is stable in repetitive trials.

Table 5: H-score of UAN + SSL with error bars on Office31.

| | A2D | A2W | D2A | D2W | W2A | W2D |
|---|---|---|---|---|---|---|
| UAN + SSL | $75.6 \pm 0.7$ | $85.4 \pm 2.6$ | $87.1 \pm 2.6$ | $76.3 \pm 1.4$ | $72.9 \pm 1.3$ | $80.0 \pm 2.5$ |

### A.3   SENSITIVITY OF HYPERPARAMETERS

We evaluated the sensitivity of the weighted hyperparameter $\alpha$ by experimenting values between 0.3 and 0.7. Figure 7 demonstrates minimal sensitivity to this hyperparameter across three settings in the Office-Home. The evaluations are conducted using UAN+SSL.

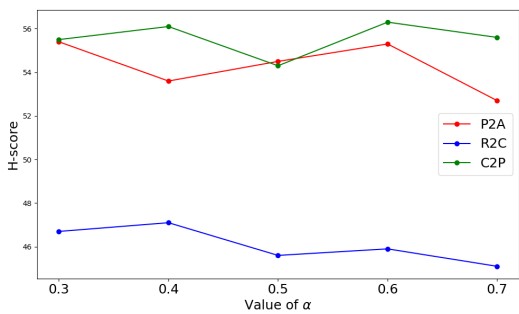

Figure 7: **Sensitivity of $\alpha$.**

Table 6: **Comparison of general and extreme settings across datasets**. The general UniDA setting refers to the conventional setup used in prior works.

| Dataset | General | | | | Extreme | | | |
|---|---|---|---|---|---|---|---|---|
| | $|\overline{\mathcal{C}}_s|$ | $\mathcal{C}$ | $|\overline{\mathcal{C}}_t|$ | SPCR | $|\overline{\mathcal{C}}_s|$ | $\mathcal{C}$ | $|\overline{\mathcal{C}}_t|$ | SPCR |
| Office-31 (Saenko et al., 2010) | 10 | 10 | 11 | 1 | 24 | 5 | 3 | 24/5 |
| Office-Home (Venkateswara et al., 2017) | 5 | 10 | 50 | 1/2 | 50 | 10 | 5 | 5 |
| Visda (Peng et al., 2017) | 3 | 6 | 3 | 1/2 | 8 | 2 | 2 | 4 |
| DomainNet (Peng et al., 2019) | 50 | 150 | 145 | 1/3 | 250 | 50 | 45 | 5 |

# B DETAILS OF EXPERIMENTAL SETUP

## B.1 EXTREME UNIDA SETTING

In Table 6, we provide the details of class distributions for our extreme settings. Following prior work, the classes in each class set are first sorted alphabetically and then divided into three groups: source-private, common, and target-private.

## B.2 METRICS

H-score (Fu et al., 2020) is defined as the the harmonic mean of accuracy on common classes $a_{\mathcal{C}}$ and accuracy on target-private (unknown) classes $a_{\overline{\mathcal{C}}_t}$.

$$\text{H-score} = 2 \cdot \frac{a_{\mathcal{C}} \cdot a_{\overline{\mathcal{C}}_t}}{a_{\mathcal{C}} + a_{\overline{\mathcal{C}}_t}}.$$

## B.3 DATASET

Office31 (Saenko et al., 2010) contains 31 classes and three domains: Amazon (A), DSLR (D), and Webcam (W), with a total of about 4k images. Office-Home (Venkateswara et al., 2017) has 65 classes and four domains: Art (A), Product (Pr), Clipart (Cl), and Realworld (Rw), with approximately 15k images. VisDA (Peng et al., 2017) is a larger dataset with 12 classes from two domains: Synthetic and Real images, totaling around 280k images. DomainNet (Peng et al., 2019), the largest DA dataset, has 345 classes and six domains, with about 0.6 million images. Following prior works (Fu et al., 2020; Chang et al., 2022; Kundu et al., 2022), we use only three domains: Real (R), Sketch (S), and Painting (P).

## B.4 IMPLEMENTATION DETAILS

We use ResNet-50 (He et al., 2016) as the backbone model for all experiments, which is pre-trained on ImageNet (Deng et al., 2009). The optimizer, scheduler and learning rate are consistent with You et al. (2019). The training steps are 10K for all experiments and the batch size is set to 36 for

both domains. The hyperparameters are set as follows: $\lambda = 0.5$ and $\alpha = 0.5$ for Office-Home, DomainNet and VisDA, and $\alpha = 0.2$ for Office31. We use SimSiam (Chen & He, 2021) as our self-supervised loss as it does not require negative samples or large batch size. The data augmentation strategy follows the same setup as SimSiam.

### B.5 CALCULATION OF UNCERTAINTY MEASUREMENTS

Let $p(y|\mathbf{x})$ represent the predicted probability distribution over the possible classes $y$ given an input $\mathbf{x}$. Specifically, $p(y_i|\mathbf{x})$ is the probability assigned to class $y_i$ for the input $x$, where $i = 1, 2, \cdots, K$ and $K$ is the number of classes. In our cases, $K = |\mathcal{C}_s|$.

**Entropy.** The entropy $H(p)$ is defined as:

$$H(p) = -\sum_{i=1}^{K} p(y_i|\mathbf{x}) \log p(y_i|\mathbf{x}) \tag{7}$$

**Confidence.** The confidence $C(\mathbf{x})$ is defined as the predicted probability for the most likely class:

$$C(\mathbf{x}) = \max_i p(y_i|\mathbf{x}) \tag{8}$$

**Energy Score.** The energy score $E(\mathbf{x})$ is calculated as:

$$E(\mathbf{x}) = -\log \sum_{i=1}^{K} \exp(p(y_i|\mathbf{x})) \tag{9}$$

**Distance.** In universal domain adaptation, source-common classes are expected to be closer to target-common classes compared to target-private classes. Therefore, we can leverage this relationship to distinguish between the different class sets. In this method, clustering is first performed on the source data, and the distance from a given input $\mathbf{x}$ to the nearest cluster centroid is used to calculate the uncertainty. Let $C_j$ represent the centroid of the $j$-th cluster, and the uncertainty score $U(\mathbf{x})$ is computed as:

$$U(\mathbf{x}) = \min_j d(x, C_j), \tag{10}$$

where $d$ is a distance metric, such as Euclidean distance. The same process can be applied when the input is from the source domain. Note that the score is updated every $k$ steps, as calculate the distances in every step is costly.

### B.6 DETAILS OF NOISE RATE EXPERIMENTS

In Figure 3, the left figures compare the misclassification rate of partial domain alignment at different noise rates against the source-only baseline. We simulate partial alignment by setting $w_\bullet(\mathbf{x}) = 1$ for common-class data and $w_\bullet(\mathbf{x}) = 0$ for private-class data in a batch, and we introduce noise by flipping these values at varying rates. The right figures display the actual average noise rates for different partial domain alignment methods.

