# OpenReview forum: "Reducing Bias in Feature Extractors for Extreme Universal Domain Adaptation"
_ICLR.cc/2025/Conference — ICLR 2025 Conference Withdrawn Submission_

### Official Review · Reviewer_paz6 · 2024-10-26

**Soundness:** 2
**Presentation:** 2
**Contribution:** 2
**Rating:** 3
**Confidence:** 4

**Summary:**

This article introduces a new setting for Extreme UniDA, aiming for UniDA methods to perform well even when the source domain has significantly more non-overlapping classes than overlapping ones. The authors analyze current methods and demonstrate their inability to achieve excellent results in such scenarios. They propose the use of self-supervised learning to explore domain-specific private knowledge to eliminate bias and enhance knowledge transfer. Experiments validate the effectiveness of this approach.

**Strengths:**

1. The problem and new setting proposed in this paper are practically significant and more aligned with real-world scenarios.
2. The motivation of the paper is supported by extensive experimental analysis, providing detailed and convincing evidence.

**Weaknesses:**

1. Lack of novelty: The fundamental issues discussed in the paper revolve around how to explore private knowledge of the target domain in an unsupervised manner free from the bias of source knowledge, and how to explore when there is no corresponding source domain knowledge for target domain-specific knowledge. However, these are not new problems, and the proposed solutions are not novel. The use of self-supervised learning to explore domain-specific private knowledge has already been discussed in [1][6], and the issue of distancing unique target domain classes from common source domain classes has been explored in [2].
2. The methods compared are too outdated; both the new setting proposed by the author and the general UniDA setting only compare work from 2022 and earlier, completely omitting many more recent studies like [3][4][5].
3. Lack of ablation study: Although only a self-supervised loss function is proposed, it is not compared with other self-supervised learning methods, which makes it difficult to provide convincing evidence.

[1] Sun Y, Tzeng E, Darrell T, et al. Unsupervised domain adaptation through self-supervision[J]. arXiv preprint arXiv:1909.11825, 2019.

[2] Hu J, Tuo H, Wang C, et al. Discriminative partial domain adversarial network[C]//Computer Vision–ECCV 2020: 16th European Conference, Glasgow, UK, August 23–28, 2020, Proceedings, Part XXVII 16. Springer International Publishing, 2020: 632-648.

[3] Zhu D, Li Y, Yuan J, et al. Universal domain adaptation via compressive attention matching[C]//Proceedings of the IEEE/CVF International Conference on Computer Vision. 2023: 6974-6985

[4] Zhu D, Li Y, Shao Y, et al. Generalized universal domain adaptation with generative flow networks[C]//Proceedings of the 31st ACM International Conference on Multimedia. 2023: 8304-8315.

[5] Qu S, Zou T, He L, et al. Lead: Learning decomposition for source-free universal domain adaptation[C]//Proceedings of the IEEE/CVF Conference on Computer Vision and Pattern Recognition. 2024: 23334-23343.

[6] Liang J, Hu D, Wang Y, et al. Source data-absent unsupervised domain adaptation through hypothesis transfer and labeling transfer[J]. IEEE Transactions on Pattern Analysis and Machine Intelligence, 2021, 44(11): 8602-8617

**Questions:**

1. I hope the authors can engage in a discussion with the paper I mentioned and further demonstrate the novelty of the proposed method.

2. I would like the authors to compare propsoed approach with more advanced methods across multiple settings to demonstrate the performance of their method.

3. Why are experiments not conducted in other settings such as PDA/OSDA/SDA?

4. Please explain the performance differences between the proposed SSL method and other SSL methods (such as rotation/cropping) in these settings, and provide more evidence to demonstrate the superiority of the proposed method.

---

> ### Author Response · Authors · 2024-11-21
>
> We thank the reviewer for the detailed and constructive feedback. We address the comments below.
>
> 1. **Novelty**
>
>
> Thank you for pointing out related works to better clarify our novelty and contribution. Please refer to General Response for details.
>
>
> 2. **Use of old baselines**
>
> We appreciate the reviewers for suggesting up-to-date baselines. We have included the latest method, MLNet, in our experiments. Please see the General Response for details.
>
> For [4] and [5], we believe that their settings and methods differ from the traditional UniDA task we focus on. Specifically, [4] explores generalized UniDA with baselines primarily incorporating active learning modules, while [5] addresses source-free UniDA, where source data is unavailable during adaptation. Nonetheless, we recognize that these settings align with our overarching aim to investigate challenging or less-explored scenarios in current research. Therefore, we have included discussions of these works in the Related Works section. For details, please refer to the updated PDF, where changes are marked in blue. For [3], we did not include it in the current version due to differences in architectures and time constraints. However, we believe that including the latest method, MLNet [7], is sufficient to demonstrate the effectiveness of our approach.
>
>
> 3. **Experiments about other settings (e.g., PDA, OSDA, SDA)**
>
> UniDA is the generalized version of the mentioned settings, which provides a comprehensive evaluation framework. Existing works [3, 8] also evaluate only on the UniDA setting. However, we agree that focusing on a specific setting can help identify the bottlenecks of different methods. In fact, our experiments across varying SPCR rates effectively represent different settings: a low SPCR closely resembles OSDA, while an SPCR greater than 1 is similar to evaluating PDA. Therefore, we think our experiments on a wide range of SPCR are enough to show the robustness of the mentioned settings.
>
> 4. **Ablation study on different SSL methods**
>
> We agree that the ablation study on different SSL methods is essential to demonstrate SSL’s  effectiveness. We are in the process of running these experiments and will update you once finished.
>
>
> [7] Lu et al. MLNet: Mutual Learning Network with Neighborhood Invariance for Universal Domain Adaptation. AAAI 2024.
>
> [8] Chang et al. Unified Optimal Transport Framework for Universal Domain Adaptation. NeurIPS 2022.

---

### Official Review · Reviewer_3xrT · 2024-10-28

**Soundness:** 3
**Presentation:** 2
**Contribution:** 2
**Rating:** 5
**Confidence:** 5

**Summary:**

This paper investigates the problem of Extreme Universal Domain Adaptation (Extreme UniDA), where the source domain contains significantly more non-overlapping classes than overlapping ones. To address this challenge, an in-depth analysis is conducted to understand why the widely-used partial domain alignment paradigm fails in the Extreme UniDA setting. Subsequently, a self-supervised learning plugin is proposed to regularize the feature extractor. Extensive experimental results demonstrate the effectiveness of this mechanism.

**Strengths:**

1.	The Extreme UniDA setting is new, and the mechanism sounds reasonable.

2.	The experimental results appear highly effective.

**Weaknesses:**

1.	In Figure 1, which specific methods do “Adversarial Based” and “Optimal Transport Based” refer to? Which dataset did you use?

2.	In section 2.2, which dataset did you use to obtain the results plotted in Figure 4?

3.	In Figure 2, what are the directions of $e\_1$ and $e\_2$, respectively? Also, I find it difficult to understand what Figure 2 conveys, could you clarify it further?

4.	Can the proposed SSL plugin also be applied to the methods in (Saito&Saenko, 2021; Hur et al., 2023; Lu et al., 2024)?

The paper has several areas that could benefit from improvement. Please consider refining it carefully to address the following example issues:

1.	Line 52: prior works have mainly adhered to the experimental protocols established by by You et al. (2019) -> prior works have mainly adhered to the experimental protocols established by You et al. (2019)
2.	In the titles of Figures 2 and 3, it is better to clarify the notations of $| \overline{C}_s |$, and $| C |$.
3.	Line 206: $\hat{y} (x)$ -> $\hat{y} (\mathbf{x})$

**Questions:**

See weakness for details.

---

> ### Author Response · Authors · 2024-11-21
>
> We thank the reviewer for the detailed and constructive feedback. We address the comments below.
>
>
> 1. **Clarity on experimental details in the figures**
>
> In Figure 1, the adversarial-based method corresponds to UAN, while the OT-based method corresponds to UniOT. The dataset used is Office31 (averaged over across tasks). In section 2.2, we report results on OfficeHome. We have updated the image and caption (marked in red) in the latest PDF.
>
> In Figure 2, we have provided a clearer description in the updated version. Specifically, we aim to convey the following: (1) the distortion of target representations is significantly more pronounced in extreme UniDA (SPCR=4) compared to unsupervised DA (SPCR=0), and (2) incorporating SSL can effectively mitigate this distortion. Consequently, SSL proves to be more effective in extreme UniDA scenarios. Please see the updated PDF for details.
>
> For $e_1$ and $e_2$, it means the basis of $\mathbb{R}^2$, i.e., [1,0] and [0,1]. For more information, please refer to [1], which is now cited in Section 3.3.
>
>
>
> 2. **Incorporation of SSL to other methods**
>
> We have included the latest method, MLNet (Lu et al., 2024), in our latest results. Please refer to General Response for details.
>
> 3. **Typos**
>
> Thank you for pointing out typos in our paper! We have addressed them in our latest version.
>
> [1] Liu et al. Self-supervised Learning is More Robust to Dataset Imbalance. ICLR 2022.

---

> > ### Comment · Reviewer_3xrT · 2024-11-25
> >
> > Thanks for your detailed reply, especially regarding Figure 2. I now understand the meaning conveyed by Figure 2. However, I believe it is insufficient to explain the motivation behind this paper. The main concern is that it relies on a 2D toy example rather than a real-world dataset. I suggest adopting a high-dimensional real-world dataset to better illustrate the motivation for utilizing self-supervised loss (SSL), as this concept is crucial for this paper. Indeed, In Figure 3 of [1], the dataset used is a high-dimensional dataset, with only two dimensions shown for visualization purposes, instead of relying solely on a 2D toy dataset.
> >
> > In addition, I have the following suggestions to further enhance the quality of Figure 2:
> >
> > 1.	Since Figure 2 is presented in the Introduction, it would be better to provide the full form of SPCR for clarity.
> >
> > 2.	In Figure 2 (b), the direction of the ellipse seems incorrect; should it align with the direction shown in Figure 2 (e)?
> >
> > 3.	For each plot in Figure 2, consider adding $\mathbf{e}_1$ and $\mathbf{e}_2$, similar to Figure 3 in [1], to enhance understanding.
> >
> > Finally, while I believe that the problem addressed in this paper is interesting, using SSL to tackle UDA-related problems has already been widely explored, as pointed out by other reviewers. Those facts limit the novelty of the approach, despite the author's efforts to clarify it. Moreover, the presentation still requires improvement, particularly in reorganizing the narrative logic of the paper to enhance clarity. Therefore, I am inclined to retain my original score.
> >
> > [1] Liu et al. Self-supervised Learning is More Robust to Dataset Imbalance. ICLR 2022.

---

> > > ### Author Response · Authors · 2024-11-29
> > >
> > > Thank you for your detailed feedback. We address your comments below. (**The latest updates are highlighted in pink in the PDF.**)
> > >
> > > 1. **Insufficient evidence of toy experiments**
> > >
> > > The toy experiments in Figure 2 and Section 3.3 are intended to provide an *intuitive* demonstration. We acknowledge the need for experiments on real datasets to further support our hypothesis. In response, **we have included a singular value spectrum analysis on a real dataset**, presented in Figure 6 and Section 4.3. The results indicate that (1) target features are prone to dimensional collapse under Extreme UniDA, and (2) SSL can effectively mitigate this collapse. For more details, please refer to the general response.
> > >
> > > 2. **Suggestion on Figure 2**
> > >
> > > Thank you for your suggestions on Figure 2. We have added the definition of SPCR to Figure 2 as recommended. Regarding the second and third point, we were unable to update them before the PDF deadline. However, as these are minor updates, we will ensure they are addressed in the next version.
> > >
> > > 3. **Limited novelty**
> > >
> > > We recognize that the unclear scope of our paper in the initial version may have contributed to the perception of limited novelty. To address this, we have refined the scope and emphasized our contributions to SSL on DA-related works in the general response and the updated PDF. Please refer to the general response for further details.

---

> > > > ### Comment · Reviewer_3xrT · 2024-11-29
> > > >
> > > > Thanks for your detailed response. However, I believe my concerns have not been fully addressed. Also, I feel that the overall quality of the paper is not yet sufficient for publication in ICLR. Therefore, I maintain my original score.

---

> > > > > ### Author Response · Authors · 2024-11-29
> > > > >
> > > > > Thank you for the response. We would appreciate it if you could spell out the key issue that we can explain/improve (for this submission, or any possible future submissions) to raise your rating on this paper.

---

> > > > > > ### Comment · Reviewer_3xrT · 2024-11-29
> > > > > >
> > > > > > Thank you for your follow-up inquiry. Please refer to my previous comments for further details. I believe there is still room for improvement, and I hope that the paper can be further refined.

---

### Official Review · Reviewer_y4ju · 2024-10-28

**Soundness:** 2
**Presentation:** 3
**Contribution:** 2
**Rating:** 5
**Confidence:** 3

**Summary:**

This paper underlines Extreme UniDA, a challenging sub-task of UniDA and illustrate that the difficulty of the task roots in the bias in the feature extractor. However state-of-the-art UniDA methods, mostly designed by partial domain alignment that removes irrelevant data by reweighting, cannot completely mitigate the bias on their own for Extreme UniDA. This paper utilizes self supervised learning to enrich the representation with the structural information of the source and target data. Extensive experiments verify that the proposed methodology effectively improves existing partial domain alignment methods across Extreme UniDA settings.

**Strengths:**

1). This paper underlines Extreme UniDA, a challenging sub-task of UniDA and illustrate that the difficulty of the task roots in the bias in the feature extractor.
2). This paper analyses the limitation of partial domain alignment across Extreme UniDA.
3). This paper proposes incorporating target label information by self-supervised learning as a lightweight module for partial domain alignment, which can reduce feature extractor bias and significantly enhance robustness across varying class-set distributions.

**Weaknesses:**

1). Supervised loss is to improve the model performance but will make the model biased to the source domain data, while self-supervised loss is to make the model learn the representation of the target domain data but will affect the classification ability of the model. How to balance these two losses .
2).The comparative work lacks the most recent research efforts, with most being from 2022 and earlier.
3). Although the method proposed in this paper has achieved a good performance on Extreme UniDA, the effect on General UniDA is not obvious, and the method lacks innovation.

**Questions:**

See above

**Details Of Ethics Concerns:**

No need

---

> ### Author Response · Authors · 2024-11-21
>
> We thank the reviewer for the detailed and constructive feedback. We address the comments below.
>
> 1. **Balance between supervised and self-supervised loss**
>
> In Section 4.3, we discuss the sensitivity of hyperparameters for self-supervised loss. The results indicate stability within the range of 0.3 to 0.7. Therefore, we think that the two losses can be effectively balanced within a reasonable range.
>
> 2. **Use of old baselines**
>
> We appreciate the reviewers for suggesting up-to-date baselines. We have included the latest method, MLNet, in our experiments. Please see the General Response for details.
>
> 3. **Generalizability of SSL**
>
> The goal of UniDA is to achieve robust performance across arbitrary label-set distributions. While existing works perform well in general UniDA, they struggle in extreme UniDA. Our objective is to highlight the limitations of current methods in extreme UniDA and propose solutions to achieve robust performance across all label-set distributions. Therefore, we respectfully disagree with the suggestion that our work lacks novelty. Instead, we argue that our study offers a novel observation: SSL primarily addresses source-private bias, which has been insufficiently explored in UniDA literature.

---

### Official Review · Reviewer_5idP · 2024-11-01

**Soundness:** 2
**Presentation:** 2
**Contribution:** 1
**Rating:** 6
**Confidence:** 3

**Summary:**

This paper addresses Universal Domain Adaptation (UniDA) in scenarios where source-private classes significantly outnumber source-common classes, termed "Extreme UniDA." The authors propose incorporating self-supervised learning ''to reduce feature extractor bias'' through a consistency loss across augmented views of target samples. The method is evaluated on four standard domain adaptation benchmarks, showing improvements over existing approaches.

**Strengths:**

- The study addresses an interesting scenario: Extreme UniDA with a high Source-Private to Source-Common Ratio.
- The integration of Self-supervised loss is easy to implement on many frameworks.

**Weaknesses:**

- Figure 1 lacks essential details such as methods, dataset, transfer tasks, and the y-axis metric used (e.g., source-private vs. target performance). Figures 2 and 3 lack discernible differences; please clarify or consider revising with more distinguishable data points. Figure 4 lacks dataset information and total class count, making the experimental setup difficult to interpret fully.

- The baselines used are old; consider including more recent baselines, such as [1] and [2].

- Although the setup is specific, self-supervision has been previously applied in UniDA, Open-set DA, and partial DA. Could the authors clarify what differentiates their self-supervision approach from that in DANCE and why this choice is preferable?

-  The title references UniDA, but the method is presented through partial DA with SSL for open-set conditions. Additionally, Figure 2 addresses an extreme partial set problem, while Figure 3 shows a vanilla domain adaptation setting. Consistency across the paper could be improved.

- The paper claims, “Given the distribution shift and label-set shift between the source data Ds and target data Dt, the learned feature extractor θf trained with Ls can become biased towards source-private classes. As a result, this bias may lead to the misclassification of target common-class data as belonging to source-private classes when evaluated on Dt.” This seems misleading. The misclassification is likely due to the dominance of source-private classes, not an inherent model bias. The larger presence of source-private classes increases the statistical likelihood of misclassification, as the model hasn’t learned any target-domain semantics. The noise experiments reinforce this point and should be highlighted more prominently.

-  SSL shows a larger improvement for adversarial methods compared to optimal transport under high SPCR. The paper could benefit from discussing why this difference occurs.

- For reproducibility, could the authors specify which classes are considered source-private, source-shared, and target-private across each dataset in the experimental setup?


[1] Liang Chen, Yihang Lou, Jianzhong He, Tao Bai, and Minghua Deng. Geometric anchor correspondence mining
with uncertainty modeling for universal domain adaptation. In CVPR, 2022

[2] LU, Yanzuo, SHEN, Meng, MA, Andy J., et al. MLNet: Mutual Learning Network with Neighborhood Invariance for Universal Domain Adaptation. In : Proceedings of the AAAI Conference on Artificial Intelligence. 2024.

**Questions:**

See Weaknesses

---

> ### Author Response · Authors · 2024-11-21
>
> We thank the reviewer for the detailed and constructive feedback. We address the comments below.
>
> 1. **Clarity on experimental details in the figures**
>
> Thank you for pointing out several unclear parts in our paper. We have made the following improvements in the updated version:
> - Figure 1: We have added details about the methods, dataset, and metrics used.
> - Figures 2 and 3 (now merged into Figure 2): The descriptions and figures have been clarified to emphasize two key points:
>     - The distortion of target representations is significantly more pronounced in extreme UniDA scenarios (SPCR=4) compared to unsupervised DA (SPCR=0).
>     - Incorporating SSL effectively mitigates this distortion, making SSL particularly effective in extreme UniDA scenarios.
> Please refer to the updated PDF for further details.
> - Figure 4 (now Figure 3): The results are based on the OfficeHome dataset with 65 classes.
>
>
> 2. **Use of old baselines**
>
> We appreciate the reviewers for suggesting up-to-date baselines. We have included the latest method, MLNet, in our experiments. Please see the General Response for details.
>
> Also, we did not include [1] in our experiments as the code is not publicly available.
>
> 3. **Difference from prior works on applying SSL to DA**
>
> Please refer to General Response for details.
>
> 4. **Consistency across settings**
>
> In Figure 2 (now Figures 2.a, 2.b, and 2.c), our goal is to focus on the target representation after training on source data. Therefore, target-private classes can be omitted for better visualization. However, we emphasize that the work remains within the scope of UniDA problems, as applying SSL to target-private classes can introduce noise—a challenge discussed in Section 3.2.
> In Figure 3 (now Figures 2.d, 2.e, and 2.f), we present the DA setting to show the distortion degree of target representations under low source-private classes.
> We have updated the descriptions in Figure 2 for better presentation.
>
> 5. **Misleading terminology**
>
> Our general idea aligns with the reviewer, however, our terminology may have been misleading and caused confusion. Our intention was to emphasize that the predictions are biased toward source-private classes, which aligns with the reviewer’s point about the increased statistical likelihood of misclassification, rather than suggesting an inherent bias in the model itself. We have made the following adjustments in our paper.
> - The title has been revised from “Reducing Bias in Feature Extractors” to “Reducing Source-Private Bias”.
> - We have clarified the paragraph in Section 2.1 (marked in orange).
>
> 6. **Difference between adversarial-based and OT-based method**
>
> Our hypothesis is that UniOT involves multiple complex modules, which may not be fully compatible. In contrast, adversarial-based methods, which rely solely on an adversarial loss design, are simpler to integrate. We leave a more comprehensive explanation for future research.

---

> > ### Comment · Reviewer_5idP · 2024-11-24
> >
> > Dear Authors,
> >
> > Thank you for your detailed response and for addressing many of the points raised in the initial review. I appreciate the effort you have put into improving the clarity and scope of the manuscript. That said, I have a few additional comments that I would like you to consider:
> >
> > -	While I acknowledge your revisions to Figure 2, I find that over-analyzing the PCA on the toy example does not add a compelling motivation. In my opinion, the figures still look quite similar, and the insights derived remain limited. I recommend conducting a similar analysis on one of the benchmark transfer tasks instead, which might better highlight the practical relevance of your approach.
> >
> > -	Thank you for updating the title and clarifying the text in lines 166–169 However, you still have in the text reference to the previous title (e.g., line 101, line 111, line 124, …).
> >
> > -	I appreciate the inclusion of MLNet in your experiments. However, could you also run MLNet as a baseline on the other datasets? I would also recommend if you have time, to add some of the methods added to the related work as baselines.
> >
> > -	The phrasing in the text (e.g., “This **discovery** further **elucidates** its application in the domain adaptation literature and explains...”) is overly strong, especially given that the claim is linked to the toy example in Figure 2. I suggest reformulating this statement to tone it down.
> >
> > Thanks

---

> > > ### Author Response · Authors · 2024-11-29
> > >
> > > Thank you for your detailed and constructive feedback. We address your comments below. (**The latest updates are highlighted in pink.**)
> > >
> > >
> > > 1. **Insufficient evidence of toy experiments**
> > >
> > > The toy experiments in Figure 2 and Section 3.3 are intended to provide an *intuitive* demonstration. We acknowledge the need for experiments on real datasets to further support our hypothesis. In response, **we have included a singular value spectrum analysis on a real dataset**, presented in Figure 6 and Section 4.3. The results indicate that (1) target features are prone to dimensional collapse under Extreme UniDA, and (2) SSL can effectively mitigate this collapse. For more details, please refer to the general response.
> > >
> > > 2. **Misleading terminology in the rest of the paper**
> > >
> > > Thank you for pointing this out! We have revised all references to “bias in the feature extractor” in the paper.
> > >
> > > 3. **Experiments on MLNet**
> > >
> > > We have run MLNet on all four datasets and updated Tables 1 and 2 accordingly. The results are consistent with our previous findings. Regarding other baselines, we were unable to run them due to time constraints. However, we respectfully believe that demonstrating improvements across three different training paradigms is sufficient to highlight the benefits of SSL.
> > >
> > > 4. **Overly strong claim on the discovery**
> > >
> > > We agree that an intuitive experiment should not be used to overly emphasize the discovery. To address this, we have included an analysis on real datasets to support our claim and corroborate the findings from the toy experiments. We respectfully think that this addition provides further insight into the application of SSL in domain adaptation. We would greatly appreciate your thoughts on whether the claim remains overly strong in light of the new results.
> > >
> > >
> > > 5. **Classes used in the Extreme UniDA setup**
> > >
> > > Thank you for pointing this out. We have reported the details in Appendix B.1 to ensure reproducibility. Specifically, following prior work, the classes in each set are first sorted alphabetically and then sequentially divided and arranged into three groups: source-private, common, and target-private.

---

> > > > ### Comment · Reviewer_5idP · 2024-12-02
> > > >
> > > > Thank you for your thoughtful responses and the updates made to the manuscript. I appreciate the effort you have invested in addressing the raised concerns, and I can see that the paper is in a better state than before.
> > > >
> > > > Regarding Figure 6, it is much clearer now, but I would suggest ensuring consistency in the color scheme (e.g., using blue for both  𝐿_{s} and 𝐿_{s} + 𝐿_{ssl}) while differentiating the lines through style changes.
> > > >
> > > > While I acknowledge the improvements, such as the inclusion of MLNet as a baseline, I still believe more work and refinement are needed. In particular, revisiting Figure 2, adding more recent baselines, and ensuring all suggested text edits are incorporated into the manuscript. Additionally, I encourage the authors to focus on making the figures more visually polished.
> > > >
> > > > Given the progress made, I am increasing my score. However, due to the areas that still require attention, I am reducing my confidence score slightly to reflect the need for additional work.

---

> > > > > ### Author Response · Authors · 2024-12-03
> > > > >
> > > > > Thank you for the positive feedback and valuable suggestions! We will continue refining the manuscript to address the remaining concerns.

---

### Official Review · Reviewer_1qWk · 2024-11-08

**Soundness:** 3
**Presentation:** 2
**Contribution:** 3
**Rating:** 6
**Confidence:** 2

**Summary:**

This paper address the challenge of Extreme Universal Domain Adaptation (UniDA), where the source dataset has many more unique classes than the target dataset. In these situations, traditional domain adaptation methods often fail because they develop a bias, focusing too much on classes unique to the source data and misclassifying target data.

To fix this, the authors propose adding self-supervised learning (SSL), which helps the model learn the structure of the target data without needing labeled target examples. By integrating SSL into current training methods, the model can better balance both datasets, reducing bias and improving accuracy, especially in difficult, high-bias cases. The results show that SSL is a simple but powerful addition that makes models perform more reliably across different class setups.

**Strengths:**

This paper addresses the difficult issue of Extreme Universal Domain Adaptation (UniDA). The source dataset has unique classes absent in the target. It introduces self-supervised learning (SSL) to reduce bias in feature extraction and also presents a novel solution. This paper have experiments on benchmarks like Office-Home and DomainNet showing consistent performance improvements. The authors clearly explain SSL's role in feature alignment, making it easy to understand. This work is relevant for real-world applications with mismatched class distributions and offer a straightforward method to enhance model robustness in high-bias situations.

**Weaknesses:**

The paper addresses limitations in existing partial domain alignment methods for extreme Universal Domain Adaptation (UniDA) but lacks novelty in its self-supervised learning (SSL) approach. The authors should clarify how their application of SSL differs from prior work and emphasize more on unique aspects that tackle class imbalance challenges.

**Questions:**

How does your application of self-supervised learning (SSL) specifically differ from existing SSL methods in the context of domain adaptation?
How many runs were conducted for the reported experiments, and what measures were taken to ensure the stability of your results?

---

> ### Author Response · Authors · 2024-11-21
>
> We thank the reviewer for the positive feedback. We address the questions below.
>
> 1. **Difference from existing SSL methods on domain adaptation**
>
> Please refer to General Response.
>
> 2. **Runs for the reported experiments**
>
> We follow most prior works (e.g., [1], [2]) on universal domain adaptation to conduct one run per experiment. Given the number of datasets and tasks in each dataset, we respectfully think it is sufficient to show the superiority and stability of our method.
>
> Furthermore, we report the results with the error bar on Office31 with three random seeds in Section A.2. We can see that the standard deviation is relatively minor compared to the performance gain of our method.
>
> [1] Lu et al. MLNet: Mutual Learning Network with Neighborhood Invariance for Universal Domain Adaptation. AAAI 2024.
>
> [2] Chang et al. Unified Optimal Transport Framework for Universal Domain Adaptation. NeurIPS 2022.

---

### Author Response · Authors · 2024-11-21

We thank all reviewers for the thorough review and constructive feedback on this work. We summarize our answers to the common questions below. We welcome additional suggestions for further improving the work.


1. **Novelty of applying SSL**

We agree that applying SSL on DA-related tasks is a concept that has been introduced previously. However, the insight that **SSL mainly addresses bias introduced by source-private classes** is a novel contribution. The observation is evident by its particular effectiveness under extreme UniDA. To the best of our knowledge, this observation has not been explored in prior works. We respectfully believe it is a significant finding that future research can build upon. To better emphasize our novelty, we have added this finding to our contribution in the Introduction section (marked in red).

We also thank the reviewers for pointing out several related works on SSL in DA-related tasks. We agree that our initial submission needed a detailed discussion to position our work better. In response, we have expanded the discussion of related works in the Related Work section (marked in red).

2. **Use of old baselines**

We appreciate the reviewers for suggesting up-to-date baselines. In response, we have included the latest method, MLNet, in our experiments. The updated results for OfficeHome and VisDA are presented in Table 2 (highlighted in red), showing improvements of 2.9% and 5.1%, respectively. We are currently running experiments on Office31 and DomainNet and will provide updates once the results are available. Preliminary findings indicate that **SSL can be effectively incorporated into the latest method.**


3. **Change of paper title**

As pointed out by Reviewer 5idP, our original title is not precise, as the bias refers to prediction bias toward source-private classes rather than inherent model bias. In response, we have updated the title to: “Reducing Source-Private Bias in Extreme Universal Domain Adaptation.”

---

### Author Response · Authors · 2024-11-29
**General Response on the scope of our work**

We sincerely thank the reviewers for their valuable feedback on the scope of our work and the intuitive experiments. After carefully considering the distinctions between our study and prior research, we have refined our contributions and scope as follows:

1. We are the first to investigate Extreme UniDA, highlighting the limitations of existing partial domain alignment methods. (Same as the original version)
2. We are the first to systematically explore various aspects of applying SSL to UniDA, including: the impact of target-private classes, the severity of source-private bias, and the benefits of combining SSL with partial domain alignment.

Below, we elaborate further on point 2.  (**The latest updates are highlighted in pink in the PDF.**)

- **The impact of target-private class**

While SSL has been studied in UDA [1] and PDA [2, 3]*, the impact of target-private classes on the effectiveness of SSL remains underexplored. In Section 3.1 and Figure 4.a, we investigate the effect of applying SSL on target-private classes and observe that while it can degrade performance slightly, the overall performance gains outweigh this drawback.

- **The severity of source-private bias**

In Section 3.3 and Figure 2, we present an intuitive experiment demonstrating that target representations become distorted under Extreme UniDA, while SSL helps preserve their structure. We acknowledge the concerns raised by Reviewers 3xrT and 5idP regarding the experiment’s limitation to a 2D dataset, which provides insufficient evidence to fully support the hypothesis. To address this, we have included a singular value spectrum analysis on a real dataset in Section 4.3 and Figure 6. The results indicate that target representations suffer from dimensional collapse [5, 6] under Extreme UniDA, and SSL effectively mitigates this issue. These findings underscore the efficacy of SSL in extreme UniDA scenarios and further validate the insights gained from our toy experiments.

- **The benefits of combining SSL with partial domain alignment**

While previous works [2, 3] have combined SSL with partial domain alignment and reported improvements, the complementary effects between the two remain poorly understood. In Section 3.2 and Figure 4.b, we demonstrate that SSL effectively reduces source-private bias and further decreases the noise rate in partial domain alignment. As a result, incorporating partial domain alignment does not introduce negative effects.



*: Notably, the “self-supervision” in DANCE [4] relies on source information, which can be influenced by source-private data. This distinction diverges from our concept of SSL and is reflected in its poor performance in our experiments.



[1] Xu et al. Self-supervised domain adaptation for computer vision tasks.

[2] Bucci et al. Tackling partial domain adaptation with self-supervision.

[3] Bucci et al. Self-supervised learning across domains.

[4] Satio et al. Universal Domain Adaptation through Self Supervision.

[5] Gao et al. Representation degeneration problem in training natural language generation models.

[6] Jing et al. Understanding Dimensional Collapse in Contrastive Self-supervised Learning.

---

> ### Author Response · Authors · 2024-11-29
> **General Response on the new experiments**
>
> We thank the reviewers for their constructive feedback on our experiments. In response, we have completed the remaining experiments on MLNet and included additional analyses to further support the findings from the toy experiments:
> 1. We ran MLNet with SSL on all four datasets, and the results consistently align with our previous findings, demonstrating that SSL reliably enhances performance (see Tables 1 and 2). We respectfully believe that the observed improvements across three different paradigms effectively demonstrate the robustness and effectiveness of SSL.
> 2. Additionally, we conducted a singular value spectrum analysis on a real dataset to further validate our hypothesis that SSL is particularly effective in extreme cases. These findings corroborate the results from our toy experiments (see Section 4.3 and Figure 6).

---

### Note · Authors · 2025-01-27

I have read and agree with the venue's withdrawal policy on behalf of myself and my co-authors.